


**Part 2: Quantitative contributions of cyanobacterial alkaline phosphatases to biogeochemical**
**rates in the subtropical North Atlantic**
*Noelle A. Held[1,2,3 *,**], Korrina Kunde[4,5,**], Clare E. Davis[6,†], Neil J. Wyatt[5], Elizabeth L. Mann[7], E.*
*Malcolm. S. Woodward[8], Matthew McIlvin[1], Alessandro Tagliabue[6], Benjamin S. Twining[8], Claire*
*Mahaffey[6], Mak Saito[1], Maeve C. Lohan[5]*
[1]Department of Marine Chemistry and Geochemistry, Woods Hole Oceanographic Institution, Woods Hole,
USA
[2]Department of Environmental Systems Science, ETH Zürich, Zürich, Switzerland
[3]Department of Biological Sciences, Marine and Environmental Biology Section, University of Southern
California, Los Angeles, CA, USA
[4]School of Oceanography, University of Washington, Seattle, USA
[5]Ocean and Earth Sciences, National Oceanography Centre, University of Southampton, Southampton, UK
[6]Department of Earth, Ocean, and Ecological Sciences, University of Liverpool, Liverpool, UK
[7]Bigelow Laboratory for Ocean Sciences, East Boothbay, USA
[8]Plymouth Marine Laboratory, Plymouth, UK
*Corresponding author: N.A. Held (nheld@usc.edu)
**These authors contributed equally
[†]now at: Springer Nature, London, UK
**Abstract**
Microbial enzymes alter marine biogeochemical cycles by catalyzing chemical transformations that
bring elements into and out of particulate organic pools. These processes are often studied through
enzyme rate-based estimates and nutrient-amendment bioassays, but these approaches are limited in
their ability to resolve species-level contributions to enzymatic rates. Molecular methods including
proteomics have the potential to link the contributions of specific populations to the overall
community enzymatic rate; this is important because organisms will have distinct enzyme
characteristics, feedbacks, and responses to perturbations. Integrating molecular methods with rate
measurements can be achieved quantitatively through absolute quantitative proteomics. Here, we use
the subtropical North Atlantic as a model system to probe how absolute quantitative proteomics can
provide a more comprehensive understanding of nutrient limitation in marine environments. The
experimental system is characterized by phosphorus stress and potential metal-phosphorus co-
limitation due to dependence of the organic phosphorus scavenging enzyme alkaline phosphatase on
metal cofactors. We performed nutrient amendment incubation experiments to investigate how
alkaline phosphatase abundance and activity is affected by trace metal additions. We show that the
two most abundant picocyanobacteria, *Prochlorocccus* and *Synechococcus* are minor contributors to



total alkaline phosphatase activity as assessed by a widely used enzyme assay. This was true even
when trace metals were added, despite both species having the genetic potential to utilize both the Fe
and Zn containing enzymes, PhoX and PhoA respectively. Serendipitously, we also found that the
alkaline phosphatases responded to cobalt additions suggesting possible substitution of the metal
center by Co in natural populations of *Prochloroccocus* (substitution for Fe in PhoX) and
*Synechococcus* (substitution for Zn in PhoA). This integrated approach allows for a nuanced
interpretation of how nutrient limitation affects marine biogeochemical cycles and highlights the
benefit of building quantitative connections between rate and "-omics" based measurements.





**Introduction**
Microbial enzymes alter marine biogeochemical cycles by catalysing chemical transformations and
facilitating the movement of elements through planetary reservoirs. On one hand, enzyme
contributions from different groups of microbes can be considered collectively, for instance in rate-
based or bioassay incubation experiments where the activities of the entire microbial community are
aggregated. On the other hand, we anticipate that the enzymes of different organisms will have
different activities and responses to perturbations; this means that resolving enzyme provenance could
enhance the quantitative connection between microbial activity and biogeochemical rates (e.g. the
goals of the fledgling Biogeoscapes program (Saito et al., 2024)). "-Omics" based methods,
particularly proteomics which directly resolves protein/enzyme concentrations, can provide a window
into the relationships between microbial abundance, enzyme concentration, and biogeochemical rates.
In this work we use quantitative proteomics to constrain the relative contributions of different
microbes (*Synechococcus* and *Prochlorococcus*) to biogeochemical rates of alkaline phosphatase
activity in the oligotrophic subtropical North Atlantic gyre. In this region, primary production is
constrained by availability of dissolved inorganic nitrogen (DIN) and phosphorus (DIP), but inputs of
atmospherically derived iron (Fe) from Saharan desert dust create a niche for nitrogen fixation,
partially alleviating nitrogen limitation but driving the system to DIP depletion(Martiny et al., 2019;
Moore et al., 2013). Lack of DIP then drives a shift towards the acquisition of the abundant yet less
bioavailable dissolved organic phosphorus (DOP) by phytoplankton(Lomas et al., 2010; Mather et al.,
2008). The DOP pool includes relatively labile phosphomono- and diesters (together ~75 to 85 % of
DOP) that derive from ribonucleic acids, adenosine phosphates and phospholipids (Kolowith et al.,
2001; Young and Ingall, 2010). These compounds cannot be directly assimilated but require the
phosphate group to be cleaved from the ester moiety first. Cleaving is catalysed by a range of
hydrolytic enzymes, such as alkaline phosphatases, which are common in marine microbes, including
bacterial as well as eukaryotic phytoplankton (Dyhrman and Ruttenberg, 2006; Luo et al., 2009;
Shaked et al., 2006). Reflecting this, alkaline phosphatase activity (APA) is high across the
oligotrophic gyres(Browning et al., 2017; Davis et al., 2019; Duhamel et al., 2010; Mahaffey et al.,
2014; Wurl et al., 2013).
Alkaline phosphatase activity is commonly regulated by intracellular phosphate levels
(Santos-Beneit, 2015) and appears to be closely linked to low ambient DIP concentrations(Mahaffey
et al., 2014). However, these enzymes also have a metal dependence, as metal co-factors are involved
in the hydrolysis process at the active site. Different alkaline phosphatases exist that, while sharing
function, evolved independently and have distinct metal requirements. For example, in *Escherichia*
*coli* (*E. coli*) the alkaline phosphatase PhoA has two $Zn^{2+}$ (zinc) or $Co^{2+}$ (cobalt) ions and one $Mg^{2+}$
(magnesium) ion at each active site per homodimer(Coleman, 1992), and in *Pseudomonas fluorescens*



the monomeric alkaline phosphatase PhoX has two $Fe^{3+}$ ions and three $Ca^{3+}$ (calcium) ions(Yong et
al., 2014). The active sites of PhoA and PhoX in marine microbes have yet to be characterized but
based on sequence homology are presumed to be like these model organisms, leading to the
hypothesis that alkaline phosphatase activity to be limited by scarce Fe, Zn, or Co trace metals in the
marine environment (Lohan and Tagliabue, 2018).
Global change is predicted to intensify phosphorus stress and alter trace metal and nutrient
cycles in the ocean (Hoffmann et al., 2012; Kim et al., 2014). Throughout the North Atlantic, the
utilisation of DOP is widespread(Mather et al., 2008) and whole community rates of APA are high
compared with other oceanic regions (Duhamel et al., 2010; Mahaffey et al., 2014). At this time, it is
not known which microbes and enzyme types are responsible for bulk APA in the North Atlantic and
elsewhere. Resolving this could lead to a more quantitative understanding of how APA activity is
regulated in the modern ocean, allowing better predictions of future changes in enzyme abundance
and activity and the resulting influence on carbon export. In this study, we use field-based quantitative
proteomics to develop an inventory of alkaline phosphatase activity and to identify nutrient-related
regulatory controls on alkaline phosphatase that are distinct for different organisms. We use this as a
proof of concept for developing quantitative connections between biogeochemical rates and "-omics"
based measurements of microbial enzymes, a topic that is of interest to ongoing international efforts
to characterize ocean metabolism.
**Methods**
**Shipboard bioassays**
All samples for this study were collected on board the *RRS James Cook* during research cruise JC150
(GEOTRACES process study GApr08), on a zonal transect at 22 °N leaving Guadeloupe on June 26[th]
and arriving in Tenerife on August 12[th], 2017, with multiple stations occupied for bioassays. A
detailed description of the bioassays and analysis of environmental parameters is presented in
Mahaffey et al. (submitted as a companion to this article).
Briefly, surface seawater was collected and processed according to trace metal clean protocols
and before dawn. For each location, duplicate or triplicate 24 L polycarbonate (Nalgene) carboys were
filled and spiked with additions of Fe, Zn or Co, as detailed in Table 1. The seawater was incubated at
ambient sea surface temperature and 50 % surface light level for 48 h from dawn to dawn with a
12:12 h simulated light cycle using white daylight LED panels.
*Table 1 Bioassay details at each station, showing the types of treatments, the amount of metal added, and the number of*
*replicates per treatment for which proteomics analyses were conducted. Note that one of the three replicates of the Fe*
*addition at the Station at 31 °W (\*) was removed as an outlier from all further analysis.*

|  | Station at 54 °W | Station at 50 °W | Station at 45 °W | Station at 31 °W |
| --- | --- | --- | --- | --- |



| | | | | | |
|---|---|---|---|---|---|
| Treatment | Control | - | - | - | - |
| | Fe | + 1.0 nM | + 1.0 nM | + 1.0 nM | + 1.0 nM |
| | Zn | + 1.0 nM | + 1.0 nM | + 0.5 nM | + 1.0 nM |
| | Co | + 50 pM | + 50 pM | + 50 pM | + 20 pM |
| Replicates per treatment | | 2 | 2 | 2 | 3* |

114       After the incubation period, subsamples for proteins were collected into acid cleaned 10 L

polycarbonate carboys (Nalgene) and immediately filtered, collecting the >0.22 µm fraction on
polyethersulfone membrane filter cartridges (Millipore, Sterivex) and recording the filtered volume.
Any remaining water was pressed out with an air-filled syringe, the filtration unit was sealed with clay
and then frozen at -80 °C. This procedure was repeated for the second (and third where applicable)
replicate of each treatment.

**120 Protein extraction and digestion**

All plastics materials were washed with ethanol and dried before usage. All samples from one station
were processed together in one extraction and digestion cycle. The frozen Sterivex filter cartridges
were transported to the laboratory on ice and cut open with a tube cutter. The filters were cut out from
their holders with razor blades and placed into 2 ml microfuge tubes (Eppendorf). Following
previously established protocols(Held et al., 2020; Saito et al., 2014)Click or tap here to enter text.,
proteins were extracted in a 1 % sodium dodecyl sulfate (SDS) buffer for 15 min at 20 °C, followed
by 10 min at 95 °C for denaturation, and 1 h at 20 °C while shaking at 350 rpm. The protein extract
was then centrifuged at 13.5 rpm for 20 min, with the impurities-free supernatant collected and then
spin-concentrated for 1 h in 5 kD membrane filters (Vivaspin, GE Healthcare). Total protein
concentrations were then measured by bicinchoninic assay (BCA) (Pierce) on a Nanodrop ND-1000
spectrophotometer (ThermoScientific). Proteins were left to precipitate in a 50:50 solvent mixture of
methanol and acetone (Fisher) with 0.004 % concentrated HCl (Sigma, ACS 37 %) for 5 days at -20
°C. At the end of the precipitation period, samples were centrifuged at 13.5 rpm at 4 °C, supernatants
were removed, and the remaining protein pellets were vacuum-dried (DNA110 Savan SpeedVac,
ThermoFisher). Pellets were redissolved in 50 µl SDS buffer, and the post-precipitation total protein
concentrations were measured via a second BCA assay to assess recovery. The protein extracts were
digested with the proteolytic enzyme trypsin (1 µg per 20 µg protein; Promega #V5280) in a
polyacrylamide tube gel(Lu and Zhu, 2005). The digested samples were concentrated by vacuum
drying and stored at -20 °C until analysis. The final volume was recorded to calculate the total protein
concentration in the processed sample, typically ~1 µg µl$^{-1}$.

**141 Target protein selection**





Protein biomarkers for *Synechococcus* and *Prochlorococcus* were chosen to detect DIP stress (PstS)
and related coping mechanisms via DOP hydrolysis (PhoA and PhoX) in our samples (Table 2). PstS
is the substrate-binding protein of the high-affinity phosphate ABC (ATP-Binding Cassette)
transporter, which is upregulated under low intracellular phosphate concentrations via the *pho* regulon
and has previously been used as an indicator of DIP stress (Cox and Saito, 2013; Martiny et al., 2006;
Scanlan et al., 1993). PhoA and PhoX are the Zn/Co-dependent and Fe-dependent alkaline
phosphatases, respectively, which facilitate the acquisition of phosphorus from the DOP pool.
*Table 2 Details on the quantified peptide biomarkers that are used to represent each protein in the subsequent plots and*
*discussions. For Prochlorococcus strains, 'HL' and 'LL' refer to high-light and low-light adapted strains, respectively.*

| | Protein | Quantified peptide (amino acid sequence) | Isolate strains with this peptide |
|---|---|---|---|
| *Synechococcus* | PhoA | HYIAVALER | WH8102 (clade III) |
| | PhoX | SQAGAELFR | WH8102 (clade III) |
| | PstS | WFQELAAAGGPK | RCC307 (clade X) |
| *Prochlorococcus* | PhoA | IYVIDPSSSPALLER | MIT9311 (clade HL II)<br>MIT9312 (clade HL II)<br>MIT9314 (clade HL II) |
| | PhoX | GNLWIQTDGK | MIT9314 (clade HL II) |
| | PstS | LSGAGASFPAK | MIT9301 (clade HL II)<br>MIT9302 (clade HL II)<br>MIT9311 (clade HL II)<br>MIT9312 (clade HL II)<br>MIT9314 (clade HL II)<br>SB (clade HL II)<br>NATL1A (clade LL I)<br>NATL2A (clade LL I) |


152        The criteria for a peptide of the protein biomarker to be used for quantification were as

follows. Firstly, we attempted to minimise the presence of methionine and cysteines because they are
subject to oxidation and cause modifications of the mass-to-charge ratio (m/z) during the analyses.
Secondly, the specificity and least common ancestor of each tryptic peptide was assessed using
METATRYP ([https://metatryp.whoi.edu/](https://metatryp.whoi.edu/)) (Saunders et al., 2020). It has been demonstrated that
carefully selected tryptic peptides, screened by using tryptic peptides databases made from genome
sequences like METATRYP, can be used to identify specific proteins in mixed microbial assemblages
to the species or even sub-species (ecotype) taxonomic resolution (Saito et al., 2015). Finally, the
performance of each precursor ion was visually inspected in Skyline (MacLean et al., 2010) for peak
shape and signal to noise-ratio during uncalibrated test measurements using a target list containing
many peptides of cyanobacterial alkaline phosphatases on a subset of the incubation samples.
**Isotopically labelled standard peptides**
The absolute quantitation of the target peptides was achieved using heavy nitrogen isotope-labelled
peptide standards (Saito et al., 2020). Briefly, DNA was synthesized containing the reverse-translated



gene sequences for our target peptides interspaced with spacer sequences and ligated with a
PET30a(+) plasmid vector using the BAMHI 5' and XhoI 3 restriction sites (Novagen; obtained
through PriorityGENE, Genewiz). Different nucleotide sequences were used to encode for the spacer
(amino acid sequence: TPELFR) to avoid repetition. As per manufacturer instructions, the plasmid
was suspended in TE buffer (10 mM Tris-HCl, 1 mM ethylenediaminetetraacetic acid) to 10 ng $\mu l^{-1}$
and of this 1 $\mu l$ was added to 20 $\mu l$ competent Tuner(DE3)pLysS *E. coli* cells on ice. The cells were
heated to 42 °C for 30 sec to initiate transformation, followed by 2 min on ice. At room temperature,
80 $\mu l$ $^{15}$N-enriched (98 %, Cambridge Isotope Laboratories), kanamycin-containing (50 $\mu l$ $ml^{-1}$) SOC
medium was added, and cells were incubated for 30 min at 37 °C at 300 rpm. Subsequently, 25 $\mu l$
were transferred to pre-heated (37 °C) 50 $\mu g$ $ml^{-1}$ agar plates and incubated overnight. One colony was
added to 500 $\mu l$ $^{15}$N-enriched SOC medium containing 50 $\mu l$ $ml^{-1}$ kanamycin as a starter culture and
incubated for 3 h at 37 °C at 350 rpm. Next, 200 $\mu l$ of the starter culture were transferred into 50 ml
flat incubation flasks with 10 ml SOC medium and incubated for approximately 3 h at 37 °C and 350
rpm until the optical density at 600 nm reached 0.6. Protein production was induced by the addition of
100 mM isopropyl β-D-1- thiogalactopyranoside to the culture and incubating at 25 °C overnight.
Inclusion bodies were initially harvested using BugBuster protein extraction protocols (Novagen).
The remaining pellet containing the inclusion bodies, i.e. the insoluble protein fraction, was
resuspended in 400 $\mu l$ 6 M urea, left on the shaker table at 350 rpm at room temperature for 3 h, and
then moved to the fridge overnight. The next morning, the proteins were reduced, alkylated, and
digested with trypsin as outlined above for the bioassay samples, and stored frozen at -20 °C until use.
**Absolute protein quantitation**
To determine the absolute concentration of the peptides in the heavy peptide mixture, commercial
standard peptides of known concentration were used. In addition to the peptides of interest, a range of
tryptic peptide sequences from commercially available standards (apomyoglobin, Sigma; Pierce
Bovine Serum Albumin, ThermoFisher) were included in the original plasmid design. Using these, the
calibrated concentration of the heavy peptide mixture had a relative standard deviation of 57 %, with
the standard deviation resulting from the cross-peptide and cross-replicate variability (n=3) (Fig. S5).
Due to the lack of reference materials, the accuracy of the protein concentrations in our bioassay
samples cannot be assessed. A systematic method-focused study addressing the precision and
accuracy of these measurements as well as the development of reference materials will be essential for
using absolute quantitative proteomics in the marine environment in the future(Saito et al., 2024). The
linear performance range of each heavy peptide standard was assessed using standard curves of the
peptide mixture. Targeted proteomic measurements were made by high pressure liquid
chromatography with tandem mass spectrometry (HPLC-MS/MS) on an Orbitrap Fusion Tribrid Mass
Spectrometer (ThermoFisher). Two $\mu g$ of each sample diluted to 10 $\mu l$ in buffer B (0.1 % formic acid
in acetonitrile) was spiked with 10 fmol $\mu L^{-1}$ of the heavy peptide mixture and injected into the




Dionex nanospray HPLC system at a flow rate of 0.17 µl min⁻¹. The chromatography consisted of a
nonlinear gradient from 5 to 95 % of buffer B with the remaining concentration consisting of buffer A
(0.1 % formic acid in LC-grade $H_2O$). Precursor ($MS^1$) ions were scanned for the m/z of the heavy
peptide standards and their natural light counterparts. The mass spectrometer was run in parallel
reaction monitoring mode and only peptides included in the precursor inclusion list were selected for
fragmentation. Absolute peptide concentrations were calculated from the ratio of the peak areas of the
product ions ($MS^2$) of the heavy peptide of known concentration to the natural light peptide
(calculated in Skyline (MacLean et al., 2010)). Manual validation of peak shapes was performed for
each peptide and sample. Differences between samples with regards to filtration volume, initial
protein mass and recovery after precipitation were accounted for. Final peptide concentrations will
hereafter be used to represent corresponding protein concentrations, with the caveat that the
measurements are not able to discern active versus non-active proteins. The status of metalation and if
the protein is correctly folded or functions as a polymeric complex cannot be determined from this
method.

**Significant responses**

Changes in protein concentrations in response to metal additions were compared relative to the
unamended control treatment after 48 h. This approach accounts for any bottle effects. Due to the
unique challenges of ocean proteomics sampling and large-scale trace-metal clean bioassays, treatment
replication was limited to n = 2 at 54 °W, 50 °W and 45 °W and to n = 3 at 31 °W. Many statistical
tests assume normal distributions, which for n = 2 is not assessable. Therefore, in our case, significant
differences in protein concentrations were evaluated using a two-fold change criterion, in which the
concentrations in all replicates of the metal treatments must lie outside a two-fold change in the
average ± one standard deviation of the control to be deemed a significant response. The fold-change
in expression and in particular the two-fold change is an accepted and commonly used metric to
identify proteins that are significantly more or less expressed across different conditions(Carvalho et
al., 2008; Lundgren et al., 2010; Zhang et al., 2006).
For the biogeochemical parameters measured in the bioassays, i.e. Chl-*a*, APA and cell counts
replication was not limited to n=2 in most cases. Where n=3, ANOVA (a=0.05) followed by Tukey
posthoc tests were applied to compare the Control treatments with other treatments.

**Results**

**Biogeochemical setting**

The oligotrophic subtropical North Atlantic is marked by high deposition of Saharan desert dust,
delivering large amounts of Fe and other lithogenic trace metals to the surface ocean(Kunde et al.,



2019). During JC150, contrasting biogeochemical regimes existed in the western and eastern basin
with high-metal, low-phosphorus, low-nitrogen surface waters at the 54 °W and lower-metal, higher-
phosphorus, higher-nitrogen surface waters 31 °W Mahaffey et al. (submitted as a companion to this
article). Furthermore, *Synechococcus* was two-fold more abundant in the west than in the east, whilst
*Prochlorococcus* was more than six-fold more abundant in the east than in the west and numerically
more abundant than *Synechococcus* throughout. Overall, the stations at 50 °W and 45 °W exhibited
biogeochemical intermediates to the conditions in the east and west. The confluence of gradients in
both DIP and trace element availability, as well as clear shifts in microbial community structure,
provide a natural field laboratory to probe how environmental drivers differentially influence the
contributions of dominant microbes to whole-ecosystem enzyme activity.
*Table 3 Date, location and biogeochemical conditions at 40 m depth at the start ($t_0$) of the bioassays. Biogeochemical*
*parameters are presented as the average ± one standard deviation of replicate $t_0$ samples, except for the singlet samples of*
*DOP at Station 4 and dCo at all stations. Mixed layer depths (MLD; defined after [52]) averages over multiple days, as these*
*were not always determined on the exact day of bioassay set-up.*

| | Parameter | Station at 54 °W | Station at 50 °W | Station at 45 °W | Station at 31 °W |
|---|---|---|---|---|---|
| **General** | Date | 11th July 2017 | 15th July 2017 | 19th July 2017 | 5th August 2017 |
| | Location | 22 °N 54 °W | 22 °N 50 °W | 23 °N 45 °W | 22 °N 31 °W |
| | SST (°C) | 27 | 27 | 26 | 25 |
| | MLD (m) | 24 ± 3 (5th to 8th July) | 33 ± 1 (12th to 15th July) | 42 ± 9 (17th to 20th July) | 51 ± 8 (4th to 8th August) |
| **Macronutrients** | DIP (nM) | 3.7 ± 2.1 | 3.7 ± 1.0 | 3.4 ± 0.8 | 14 ± 0.70 |
| | DOP (nM) | 87 ± 7.5 | 137 ± 39 | 112 | 129 ± 29 |
| | DIN (nM) | 1.5 ± 1.9 | 1.66 ± 0.56 | 3.36 ± 1.0 | 6.2 ± 0.0 |
| | APA (nM h$^{-1}$) | 2.8 ± 0.21 | 2.86 | 2.48 ± 0.10 | 1.15 ± 0.08 |
| **Trace metals** | dFe (nM) | 1.26 ±0.06 | 0.53 ± 0.06 | 0.83 ± 0.00 | 0.23 ± 0.05 |
| | dZn (nM) | 0.25 ± 0.14 | 0.46 ± 0.09 | 0.14 ± 0.01 | 0.04 ± 0.01 |
| | dCo (pM) | 11.0 | 11.1 | 13.0 | 13.9 |
| **Phytoplankton community** | *Synechococcus* (cells ml$^{-1}$) | 3.4 ± 0.55 · 10$^3$ | - | - | 1.6 ± 0.26 · 10$^3$ |
| | *Prochlorococcus* (cells ml$^{-1}$) | 29 ± 0.37 · 10$^4$ | - | - | 181 ± 0.37 · 10$^4$ |
| | Chl-*a* (µg L$^{-1}$) | 0.064 ± 0.01 | 0.055 ± 0.01 | 0.110 ± 0.06 | 0.149 ± 0.005 |


**'Traditional' bioassay responses to the metal-amended bioassays**
Across the 16 bioassays conducted across the North Atlantic, spanning various metal conditions due
to natural gradients and metal amendments, the bioassays did not result in any observable responses in



the conventional parameters such as Chl-*a*, APA, and cell counts for *Prochlorococcus* and
*Synechococcus*. Following a 48-hour incubation period, there were no statistically significant changes,
either positive or negative, compared to the unamended Control (see Fig. 1). This finding remained
consistent in additional experiments conducted at the same location and time as outlined in Mahaffey
et al. (submitted as a companion to this article). At least in part, this may be due to the large
differences between replicate incubation bottles which may have resulted from stochasticity of
sampling the low biomass system of the subtropical North Atlantic. Regardless, the absence of
significant responses in the biogeochemical parameters contrasted notably with the observed
responses in protein data detailed below.

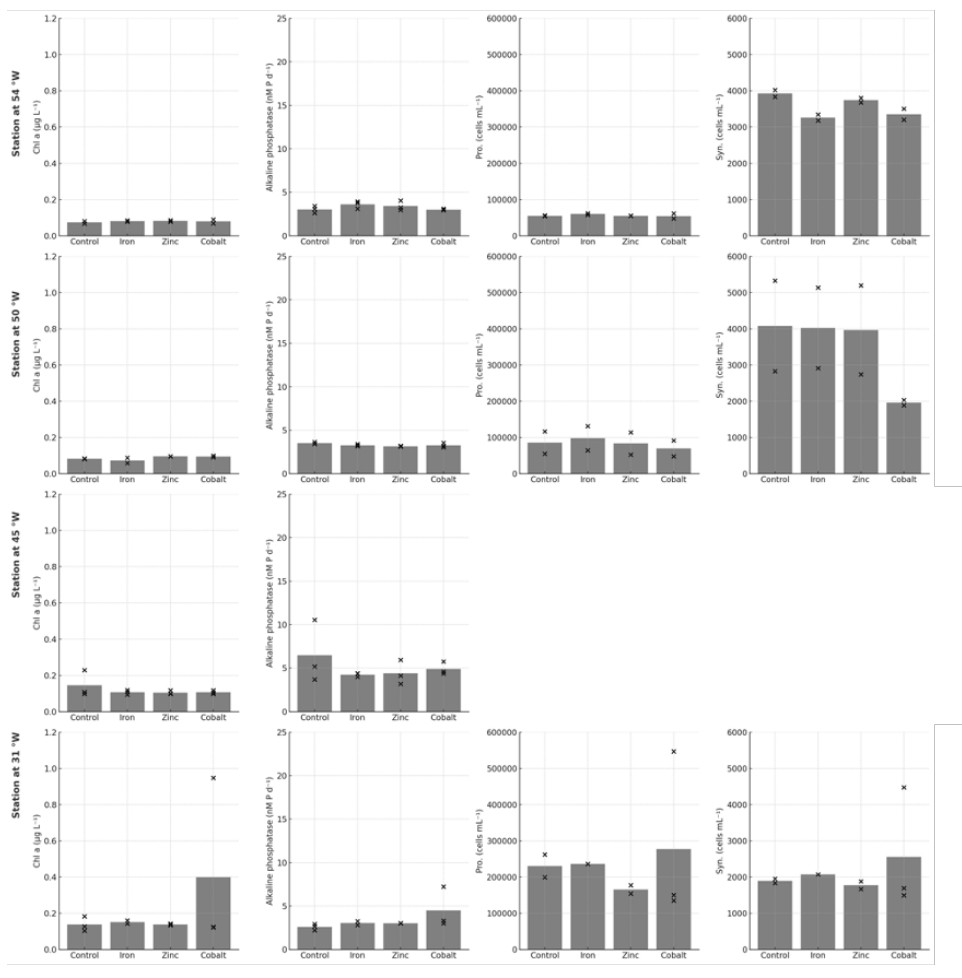


*Figure 1 Mean concentrations (grey bars) of the bioassay parameters after the addition of Fe, Zn and Co at the four*
*stations), specifically (left column) concentrations of chlorophyll a (µg L⁻¹), (second to left column) rates of alkaline*
*phosphatase (nM d⁻¹), (second to right column)* Prochlorococcus *abundance (cells mL⁻¹) and (right column)* Synechococcus



*abundance (cells mL⁻¹). Dots represent the concentrations of each replicate. Note the data gap for cell counts at the Station*
*at 45 °W.*

**268 Absolute concentrations of strain-resolved cyanobacterial alkaline phosphatases**

In contrast to the bioassay results, there were clear changes in organism-resolved alkaline phosphatase
concentrations after metal additions. We focused on the enzymes PhoA and PhoX and used peptides
that were specific to one or more strains of either *Prochlorococcus* or *Synechococcus* (Table 2) and
represent a subset of the population of alkaline phosphatase enzymes in the ocean. We note that
marine alkaline phosphatases are found at different subcellular localizations and are also known to be
secreted to the environment (i.e. into the dissolved phase) (Li et al., 1998; Luo et al., 2009). Our
measurements focus on the alkaline phosphatase associated with microbial cells, i.e. the particulate
phase. Coming from an overview of the enzyme concentrations across isoforms, taxa and bioassays,
we will discuss how these compare to the APA assay involving fluorogenic substrates.

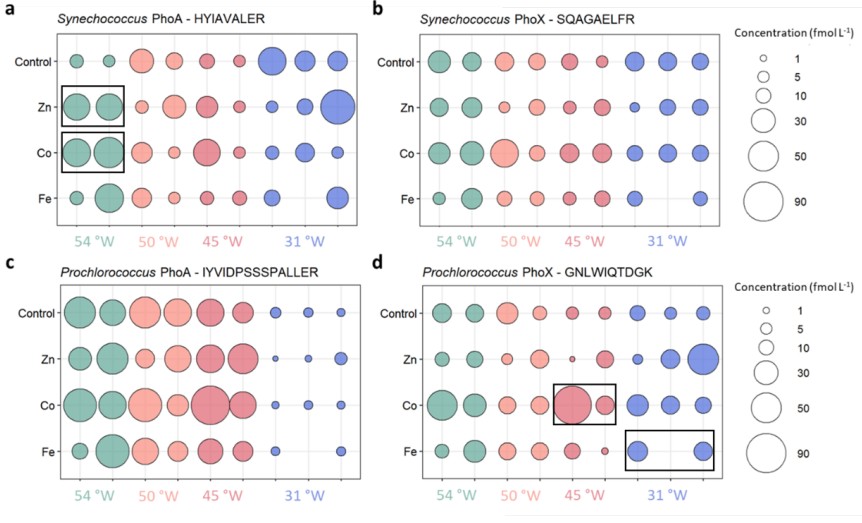


*Figure 2 Absolute concentrations of the alkaline phosphatases PhoA (left column) and PhoX (right column) of*
*Synechococcus (top) and Prochlorococcus (bottom) in the different metal treatments or the unamended control at the four*
*probed stations. Bubbles of the same colour are replicates of the same treatment and show the concentrations as fmol*
*enzyme per L seawater. Black boxes indicate significant change from Control treatment.*
The results of all measured alkaline phosphatase concentrations are shown in Fig. 2 and all
data is compiled in Table S3. *Synechococcus* PhoA and PhoX concentrations in the control treatments
ranged from 6 to 43 fmol L⁻¹ and 6 to 26 fmol L⁻¹, respectively, with no clear cross-basin trend despite
a strong west-to-east decreasing gradient in *Synechococcus* cell abundance (Table 3). Similarly,
*Prochlorococcu*s PhoA and PhoX concentrations in the control treatments ranged from 2 to 55 fmol
L⁻¹ and 6 to 23 fmol L⁻¹, respectively, but with elevated PhoA at in the west and the lowest
concentrations at 31 °W, which is opposite to the west-to-east increasing gradient in *Prochlorococcus*



cell abundance. This suggests that we observed a gradient in DIP/trace metal nutrient stress for
*Prochlorococcus*, but not for *Synechococcus*.
Our measured alkaline phosphatase concentrations were similar, albeit at the lower end, to
concentrations reported for other cyanobacterial enzymes and nutrient regulators from the North
Pacific ($\sim 10^{-1}$ to $10^3$ fmol $L^{-1}$) (Saito et al., 2014). Interestingly, our alkaline phosphatase
concentrations occurred at the same concentration range as other macronutrient stress indicators
(response regulator protein PhoP, sulfolipid biosynthesis protein SqdB, nitrogen regulatory protein P-
II), all of which did not exceed tens of fmol $L^{-1}$ (Saito et al., 2014). In contrast, concentrations of the
*Prochlorococcus* PstS transporter protein were higher, ranging from 95 to 472 fmol $L^{-1}$ (Table S3).
This is within the concentration range of other cyanobacterial nutrient transporters, such as the urea
transporter UrtA, measured previously (Saito et al., 2014). In mediating nutrient stress, particularly
phosphorus stress, the relative role of transporter proteins (such as PstS) versus other strategically
deployed enzymes like alkaline phosphatase in the oligotrophic specialists *Synechococcus* and
*Prochlorococcus*, represents an interesting avenue for future research.

**Metal control on alkaline phosphatases**

Our strain-specific, quantitative proteomics approach allowed us to resolve contrasting responses
across the sites. The responses differed with varying phytoplankton species, alkaline phosphatase
form and stimulating metal addition, consistent with differences in the biogeochemical regimes (Table
3). At the iron-rich westernmost station, the *Synechococcus* PhoA concentration increased six- and
seven-fold upon addition of Zn (to $38 \pm 0.56$ fmol $L^{-1}$) and Co (to $47 \pm 6.8$ fmol $L^{-1}$) relative to the
control ($6.7 \pm 1.5$ fmol $L^{-1}$), respectively. At one intermediate Station (45 °W), the *Prochlorococcus*
PhoX concentration increased 8-fold upon addition of Co relative to the control. Notably, a direct
response of alkaline phosphatase to an addition of Co has not been shown in the field before. In
contrast, at the low iron easternmost station, the *Prochlorococcus* PhoX increased over two-fold upon
Fe addition (to $18 \pm 2.6$ fmol $L^{-1}$) relative to the control ($8.2 \pm 2.4$ fmol $L^{-1}$).
At least three scenarios are possible to explain the increased alkaline phosphatase
concentrations of *Synechococcus* and *Prochlorococcus* in seawater in these treatments – two
biochemical and one growth driven hypotheses. First, the metal addition may stimulate the production
of the alkaline phosphatase enzyme via a direct or indirect metal regulation on the expression of this
enzyme, as was previously observed for PhoA with Zn additions in *Synechococcus* cultures (Cox and
Saito, 2013). Second, the metal addition may prevent the degradation of the existing alkaline
phosphatases by filling empty metal co-factor sites (Bicknell et al., 1985), with the caveat that PhoA
is likely to be periplasmic and hence unlikely to be actively degraded (Luo et al., 2009). Both
biochemical scenarios allow for increased alkaline phosphatase concentrations at a constant cell



abundance. The third explanation is that the alkaline phosphatase concentration increases because the metal addition stimulates overall cell growth, resulting in higher phosphorus demands and hence more production of alkaline phosphatase proteins by the cell. This could manifest itself as higher cell abundances in addition to increased alkaline phosphatase concentration per unit biomass.

While the different scenarios are not mutually exclusive, our quantitative proteomic approach allowed us to discern between biochemical and growth mechanisms by normalising the alkaline phosphatase concentrations to the total cell counts of *Prochlorococcus* and *Synechococcus*, caveating that the cell counts are not strain-specific, unlike the peptide-based protein measurements. Cell counts did not change significantly across these treatments Mahaffey et al. (submitted as a companion to this article). which means that the trends of increased alkaline phosphatase concentration per L seawater persisted in bioassays (i.e. +Zn and +Co at 54 °W and +Fe at 31 °W; cell counts do not exist for 45 °W) even when converted to the number of alkaline phosphatase enzymes per cell, indicating biochemical regulation as opposed to simply growth of the responsible organism. Specifically, the concentration of *Synechococcus* PhoA increased to $8418 \pm 673$ enzymes cell[-1] upon Co addition and to $6057 \pm 48$ enzymes cell[-1] upon Zn addition relative to $1025 \pm 257$ enzymes cell[-1] in the control at 54 °W, while the concentration of the *Prochlorococcus* PhoX increased to 59 enzymes cell[-1] upon Fe addition relative to $19 \pm 7$ enzymes cell[-1] in the control at 31 °W. Therefore, a direct biochemical metal control on the alkaline phosphatase concentrations during the bioassays is plausible (i.e. either of the first two explanations) and adds weight to the hypothesis for the localised metal-phosphorus co-limitation in the subtropical North Atlantic (Browning et al., 2017; Jakuba, R. Wisniewski et al., 2008; Mahaffey et al., 2014; Saito et al., 2017; Shaked et al., 2006).

These estimates of enzyme copies per cell are potentially underestimates as multiple *Prochlorococcus* and *Synechococcus* ecotypes co-exist and the alkaline phosphatase peptide sequences probed here do not encompass all of them (Table 2). Moreover, it is also possible that there are additional isoforms of alkaline phosphatase present in these organisms that have yet to be identified. Yet in these marine cyanobacteria, the cellular concentration of alkaline phosphatase was much higher compared to a measurement in the model bacterium *E. coli*, which contained ~4 PhoA copies cell[-1] (Wiśniewski and Rakus, 2014). This underscores the ecological demand for alkaline phosphatases due to the significant depletion of phosphorus in the marine environment. It is yet to be determined whether the per-cell estimates of alkaline phosphatases presented here are the norm for marine cyanobacteria, or whether these estimates are exceptionally high due to the prevalence of phosphorus stress in our study region.

While PhoX enzymes are unknown to use Co as a metal co-factor and the response at 45 °W warrants further investigation, the substitution of Zn with Co in PhoA has been hypothesised previously based on the distributions of trace metals and phosphate in the Sargasso Sea (Jakuba, R.



Wisniewski et al., 2008; Saito et al., 2017). The results from 54 °W support this hypothesis as the
addition of both Zn and Co were associated with almost equal increases of the *Synechococcus* PhoA
concentration relative to the control. It is thought that while Zn is the preferred metal centre for PhoA,
it is possible to substitute Co for Zn in the protein, such as occurs in *Thermotoga maritima*
(Wojciechowski et al., n.d.) *in vivo* and *in vitro* in *E. coli* (Gottesman et al., 1969). Metabolic
substitution capabilities between Zn and Co in carbonic anhydrases have previously been identified in
marine phytoplankton, with similar or slightly reduced growth rates for a range of marine diatoms and
coccolithophores, when Zn was replaced with Co in carbonic anhydrases (Dupont et al., 2006;
Kellogg et al., 2020; Morel et al., 2020; Price and Morel, 1990; Sunda and Huntsman, 1995;
Timmermans et al., 2001; Xu et al., 2007; Yee and Morel, 1996). However, in certain organisms such
as in the coccolithophore *Emiliana huxleyi*, it is possible that Co is the preferred metal co-factor since
the growth rate was higher under replete Co than under replete Zn. One reason could be the co-
evolution of ocean chemistry and cyanobacteria under the Co- and Fe-replete, but Zn-deplete
conditions of the ancient ocean ~2.5 Gyr ago (Dupont et al., 2006; Johnson et al., 2024; Saito et al.,
2003). Another explanation for Co use in alkaline phosphatases may require maintaining low
intracellular availability of Zn to avoid toxicity through inhibition of cobalt insertion by high Zn into
cobalamin (Hawco and Saito, 2018). Supporting this, the cyanobacterium *Synechococcus bacillaris*
and *Prochlorococcus* were found to have absolute Co requirements for growth(Sunda and Huntsman,
1995). Together with these aspects, our insights from the bioassay response at 54 °W merits further
investigations into whether *Synechococcus* can interchange Zn and Co in PhoA, and indeed which
metal is preferred. This would be an important insight for considerations of stoichiometric plasticity
and niche partitioning across the vast Zn- and Co-depleted regions of the ocean, especially where dZn
can become depleted to levels similar or below dCo (Kellogg et al., 2020).

382       Across all bioassays, the addition of Zn did not increase the concentration of any presumably

Fe-dependent PhoX, and the addition of Fe did not increase the concentration of any presumably Zn-
or Co-dependent PhoA. In other words, no significant unexpected responses were observed.
Nevertheless, there are some non-significant trends upon the addition of Co that warrant further study.
For example, the addition of Co increased the putative Fe-containing *Prochlorococcus* PhoX protein
concentration dramatically in one of the replicates at 45 °W and hence, Co could be an efficient metal
co-factor in PhoX (as in the bacterium *Pasteurella multocida* (Wu, Jin-Ru et al., 2007)), or at least,
directly or indirectly stimulate production of PhoX. This contrasts with  the results of Kathuria &
Martiny (Kathuria and Martiny, 2011), who hypothesized an enzyme inhibiting role of Co (and Zn;
with Fe untested) for the activity of both *Synechococcus* and *Prochlorococcus* PhoX by replacing
$Ca^{2+}$ at the active site.

393       Taken together, the results of our bioassays suggest that alkaline phosphatase enzymes are

affected by trace metal concentrations, and that the response to Zn, Co or Fe may be species or strain



specific. The metal effects differed between the responsive enzyme type (PhoA versus PhoX) and
phytoplankton species (*Prochlorococcus* versus *Synechococcus*) at contrasting biogeochemical
settings across the basin. It is plausible that the significant changes in protein concentration can result
directly from the metal addition triggering more alkaline phosphatase production per cell. This
demonstrates that the cycling of macronutrients and metals are intermittently linked and that the
nature of that linkage depends on microbial community composition

**401   Alkaline phosphatase abundances in the context of bulk community APA**

Alkaline phosphatase activity and phytoplankton biomass (by Chl-*a* proxy) did not increase
significantly upon the metal additions in the bioassays, neither together with the responses observed in
the absolute enzyme concentrations at 54 °W, 45 °W and 31 °W, nor in any other treatments or
locations (also Mahaffey et al.; submitted). A quantitative explanation for this – and hence a
demonstration of the power of proteomics on the organism level - emerges from estimates of enzyme
abundance-based enzyme rates, where the APA assay covers the entire microbial community (i.e.
everything from bacteria to eukaryotes) but the proteomics measurements are specific to a subset of
*Prochlorococcus* and *Synechococcus*.

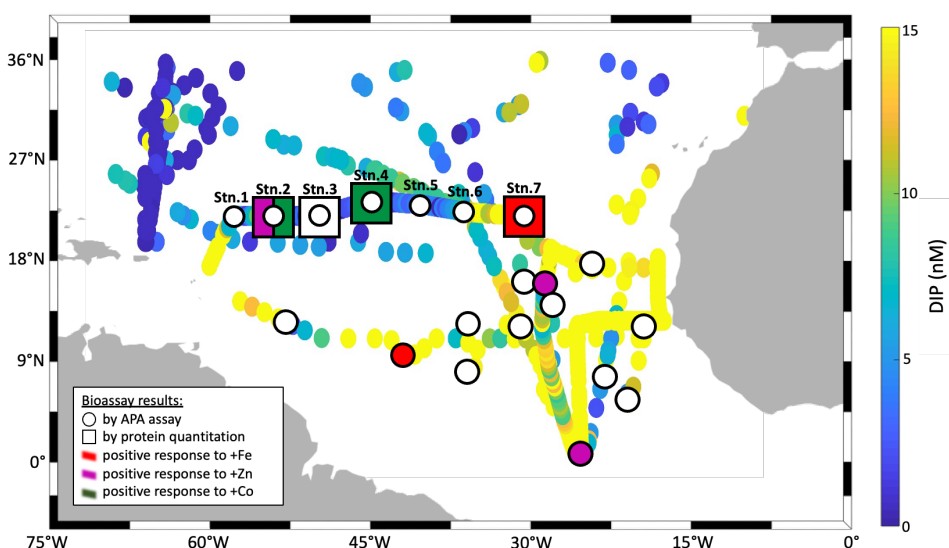


*Figure 3 Map of the North Atlantic showing surface phosphate concentrations (compiled by Martiny et al., 2019(Martiny et al., 2019); augmented with data from Browning et al. 2017(Browning et al., 2017)). Overlain are locations of bioassays, where the response of APA to metal additions was tested (circles), and of bioassays, where the absolute concentration of the alkaline phosphatase proteins was measured in response to metal additions (squares). Bioassays of the present study include longitudinal station labels. The others are from Mahaffey et al. (2014)(Mahaffey et al., 2014) and Browning et al. (2017)(Browning et al., 2017) as well as from additional bioassays during JC150 Mahaffey et al. (submitted as a companion*





*to this article). but where no protein measurements were made. Symbols at bioassay locations are coloured in red, purple or*
*green, if a positive response was observed upon addition of Fe, Zn or Co respectively.*
The geographically localised changes in the alkaline phosphatase concentrations and the
absence of changes in APA during our bioassays align with the results of Browning et al.(Browning et
al., 2017), where only one in eight experiments showed a metal driven response in APA. Furthermore,
a compilation of bioassay APA measurements from the North Atlantic exposes both the absence and
diversity in metal responses across the basin (Fig. 3). For example, Mahaffey et al.(Mahaffey et al.,
2014) observed a positive response of APA to Zn in the eastern basin, whereas our bioassay results
showed an increase in *Synechococcus* PhoA concentration (but no change in APA) upon Zn addition
in the western basin. While this difference is potentially the result of regional and seasonal drivers,
resolving the apparent 'patchiness' of these trends will rely on the better spatial and temporal
coverage of similar studies in the future. One explanation for the presence of the many null responses
across the basin is that organisms effectively re-allocate metals towards use in alkaline phosphatases
when under phosphorus stress. A comparable re-allocation mechanism of cellular Fe between
metalloproteins involved in biological $N_2$ fixation and photosynthesis has previously been
demonstrated in the diel cycle of *Crocosphaera watsonii* (Saito et al., 2011b).
**Towards a quantitative, in situ marine metalloproteome of *Synechococcus***
An advantage of absolute quantitative measurements over relative proteomics data is the ability to
relate the absolute protein concentrations to other data types, including biological rate measurements
and cellular metal stoichiometry. To this end, the concentrations of the Zn, Co or Fe-dependent
alkaline phosphatases measured in this study naturally lead to two questions: First, how much metal is
allocated as alkaline phosphatase co-factors in the cell, and how does this compare to total cellular
metal content? Second, how does the APA estimated from enzyme abundance compare to assay-based
APA?
To address these questions, model calculations were performed. Variables other than the
absolute concentrations of the alkaline phosphatases were either measured concomitantly during the
bioassays, such as cellular metal quotas, cell abundance, APA and DOP concentration, or sourced
from the literature such as strain specific contribution to cell abundance, phosphoester contribution to
the DOP pool, enzyme kinetics parameters and subcellular enzyme localization. We chose to use the
*Synechococcus* PhoA concentrations in the control treatments at 54 °W after 48 h in these calculations
for three reasons: First, cellular metal quotas of *Synechococcus* but not of *Prochlorococcus* were
measured in this treatment, due to limited sampling capacity. Second, enzyme kinetics parameters of
the PhoA rather than PhoX isoform are well documented in the literature(Lazdunski and Lazdunski,
1969). Third, estimates for the contribution of *Synechococcus* strain WH8102 (to which our measured
PhoA is specific) to total *Synechococcus* counts exist from previous studies nearest to 54



°W(Ohnemus et al., 2016). A similar reasoning applied to the phosphoester contribution to the DOP
pool. For ease, more detailed explanations, all values, and assumptions are in Tables S1 and S2.
Equation 1a approximates the cellular Zn allocation towards the *Synechococcus* PhoA from
the replicate-averaged PhoA concentration in seawater normalised to cell abundance, assuming full
metalation of the enzyme with four metal ions per dimer(Coleman, 1992). The cell abundance is a
function of *Synechococcus* cell counts and the fractional abundance of strain WH8102, to which the
measured PhoA is specific. Equation 1b expresses the results of Equation 1a as a fraction of the total
cellular Zn content.
Allocated $Zn_{PhoA}$ = metalated co-factors$_{PhoA}$ * PhoA$_{SW}$ / (*Syn.* abundance * WH8102 fraction)   **(1a)**
Fractional allocated $Zn_{PhoA}$ = $Zn_{PhoA}$ / $Zn_{total\ cell.}$   **(1b)**
The amount of metal allocated to PhoA in *Synechococcus* is 3,054 atoms cell$^{-1}$. This translates
to a maximum fractional contribution towards the total cellular Zn content of 0.66 % after dividing by
cellular Zn measured using SXRF (Table S1). If the co-factor in PhoA is assumed to be occupied by
$Co^{2+}$ instead of $Zn^{2+}$, the fractional contribution to the cellular Co content is 38 %, due to the lower
total cellular Co content of *Synechococcus* compared to Zn (Table S1). It is possible that the active
sites of PhoA are occupied by a mixture of Zn and Co, incompletely metalated, or under competition
by metals other than Zn or Co. Nevertheless, these low fractional contributions of PhoA-allocated Zn
appear biochemically reasonable, as the majority of Zn in *Synechococcus* appears to be stored in
metallothioneins to maintain Zn homeostasis and potentially supply alkaline phosphatases with Zn as
needed (Cox and Saito, 2013; Mikhaylina et al., 2022). However, our bioassay results also suggest Zn
may not be the preferred co-factor in *Synechococcus* PhoA: The larger response of this enzyme to Co
additions at 54 °W suggests the effective substitution of or even preference for Co (Fig. 2). This
would align with evolutionary arguments (see '*Metal control on alkaline phosphatases*') and imply
that PhoA is a potential major sink of cellular Co. This would also imply that *Synechococcus* growth
may be sensitive towards Co-phosphorus co-limitation in the oligotrophic ocean.
Equation 2a approximates the *Synechococcus* PhoA-abundance based hydrolysis rate as a
function of the PhoA concentration (converting molarity units to grams using its molecular weight),
phosphoester substrate concentration, and Michaelis-Menten kinetics parameters $V_{max}$ and $K_m$, the
maximum reaction rate and half-saturation constant, respectively, derived from an *E. coli* homologue
of PhoA (Table S2). Equation 2b expresses the results of Equation 2a as a fraction of the 'total APA',
a function of the measured MUF-P assay-based APA with a correction applied for the subcellular
localisation of marine alkaline phosphatases, of which only the periplasmic-outwards fraction (~20 to
80 %) is detectable via the MUF-P assay. In other words, the calculated 'total APA' accounts for both
the dissolved and particulate activity. Details on made assumptions are in the supplement (Table S2).



$\text{Rate}_{PhoA} = \text{PhoA}_{SW} * \text{molecular weight} * V_{max} * \text{substrate} / (\text{substrate} + K_m)$     **(2a)**

487         where substrate = DOP * phosphoester fraction

Fractional $\text{Rate}_{PhoA} = \text{Rate}_{PhoA} * \text{periplasmic-outwards fraction} / \text{assayed APA}$     **(2b)**
The protein abundance-based rates range from 0.00517 nM h$^{-1}$ to 0.213 nM h$^{-1}$ for the Zn-
dependent *Synechococcus* PhoA and from 0.00428 nM h$^{-1}$ to 0.0187 nM h$^{-1}$ for the Co-dependent
PhoA (Fig. 4a and b), using the *E. coli* kinetics parameters for each metal that are slower under Co
coordination. In terms of fractional contributions to total APA, the rate estimates translate to
maximally 5.2 % for Zn-PhoA and 0.46 % for the Co-PhoA. Regardless of the choice of enzyme
kinetics, it appeared that *Synechococcus* PhoA contributed a small component to the total APA in our
bioassays. This concurs with the observed increase in concentration of *Synechococcus* PhoA upon
metal addition versus the null response in APA. Applying the same calculation and kinetics
parameters for the case of the *Prochlorococcus* PhoA yields abundance-based rates between 0.0347
nM h$^{-1}$ and 1.42 nM h$^{-1}$ for the Zn-PhoA and between 0.0287 nM h$^{-1}$ and 0.125 nM h$^{-1}$ for the Co-
PhoA, which translate to maximal contributions to the total APA of 35 % and 3.1 %, respectively.
These higher rates and fractional contributions compared to the *Synechococcus* PhoA are due to the
higher concentrations of the *Prochlorococcus* PhoA than the *Synechococcus* PhoA in the chosen
samples. The taxon-specific alkaline phosphatase concentrations illustrate the challenge of
interpreting bulk enzyme activities when the functional enzyme class is produced by many biological
taxa (cyanobacteria, heterotrophic bacteria, diatoms etc.). In essence, the different bioassay responses





shown here demonstrate the need to further develop a "meta-biochemistry' capability to understand
biogeochemical reactions at the mechanistic level.

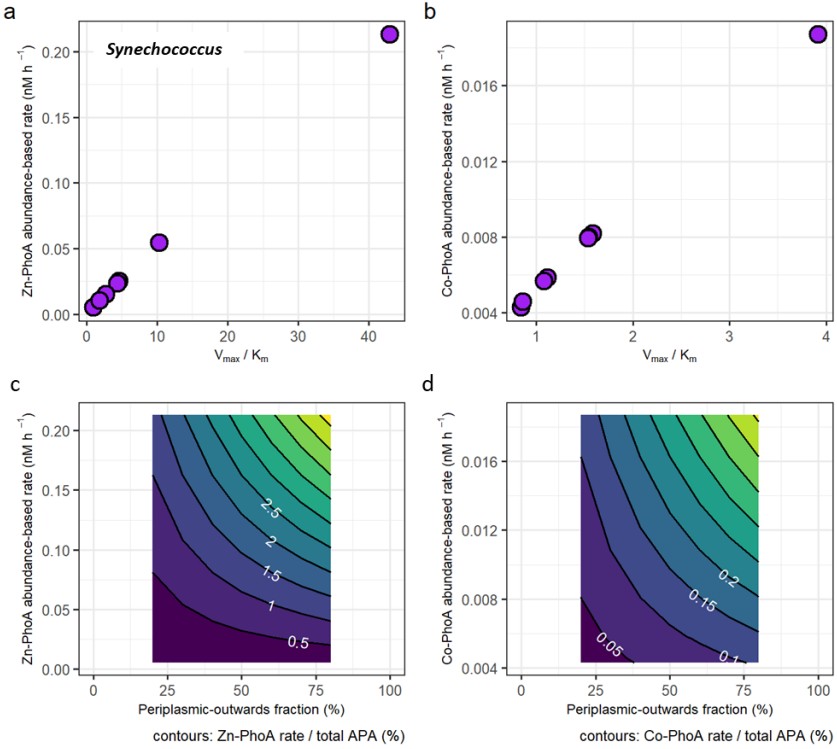


*Figure 4 (a) Protein abundance-based APA estimates of Synechococcus Zn-dependent PhoA as a function of different*
*enzyme kinetic parameters $V_{max}$ and $K_m$. (b) Same as (a), but with enzyme parameters for the less efficient Co-dependent*
*PhoA. (see Table S2). (c) The fraction of the Zn-PhoA abundance-based APA from (a) over the total APA. (d) Same as (c)*
*but using the Co-PhoA rates from (b). Note the scale difference between (a,c) and (b,d).*
The enzyme-based rates calculated here may be below that of bulk activity (APA assay) due
to our focus on a few species within the particulate phase of the enzyme. Alkaline phosphatase is
known to be more abundant in the dissolved phase, for example as much as 72% of the APA was
observed in Red Sea samples to be in the dissolved phase(Li et al., 1998). Moreover, the periplasmic
location of alkaline phosphatase has been observed to result in loss during preservation. In a
preservation study, PhoA was notably the protein with the lowest recovery in *Synechococcus*
WH8102 after a month in storage compared to 101% ± 27% for the fifty most abundant proteins(Saito
et al., 2011a). The combination of multiple abundant and rare microbial sources of alkaline
phosphatases together contribute to the particulate, and when secreted or lost, dissolved reservoirs that
make up the bulk APA.
**Conclusions**



This study performs taxon-specific alkaline phosphatase isoform analysis via absolute quantitative
proteomics on *Prochlorococcus* and *Synechococcus,* coupled to enzyme bioassays. This approach
supports the use of Zn, Co and Fe in alkaline phosphatases in the natural oceanic environment, but
also adds complexity to our understanding of how these enzymes are regulated in a biogeochemical
context. Our mechanistic perspective revealed that these two highly abundant microbes are only
minor contributors to bulk APA, which carries important implications for the interpretation of the
widely used fluorescent APA assay. Additionally, within this picocyanobacterial class, we observed
heterogeneous responses of the alkaline phosphatase enzymes depending on the protein, taxonomy,
biogeochemical context, and treatment. This indicates that there is significant biological diversity in
the responses of individual marine organisms to experimental treatments that can be resolved by
combining enzyme assay measurements with quantitative proteomics. Our results indicate a need for
biochemical characterisation of key marine alkaline phosphatases, particularly with regards to their
kinetics and metal co-factors as highlighted by the potential importance of Co as a metal co-factor in
PhoA and possibly PhoX. Future efforts to understand the biochemical properties of marine microbes
will benefit the connected interpretation of molecular, enzymatic, and biogeochemical assays, and in
turn our understanding of nutrient cycling in the ocean system.
**Acknowledgements**
The authors would like to thank the captain and crew of the *R.R.S. James Cook* during cruise JC150,
as well as all the scientific party members, who helped to conduct the bioassays. The authors would
also like to thank Alastair Lough and Clément Demasy for the dCo measurements, and Julie Robidart
and Rosalind Rickaby for fruitful discussions on this manuscript. This research was supported by the
National Environmental Research Council (UK) under grants NE/N001125/1 to MCL and
NE/N001079/1 to CM, by the National Science Foundation (USA) under grant OCE1829819 to BST,
OCE1924554, OCE1850719 and NIH R01GM135709 to MAS, and by the Graduate School of the
National Oceanography Centre Southampton (UK) to KK. The writing process was also supported by
the Simons Foundation under award 723552 to KK and by the the USC Dornsife College of Arts and
Sciences to NAH.
**Author contributions:** NAH and KK wrote the initial manuscript draft. NAH and KK performed the
proteomics analysis with help from MM and MAS. KK, NAH, NJW, CD, CM, MCL performed the
experiments at sea. BST and ELM conducted the cell quota measurements. MCL, CM, AT and MAS
led the research campaign. All authors commented on the manuscript.
**Competing interests**: The authors declare no competing interests.
**Data Availability:** Source data for all main and supplementary figures are provided in the
supplement. The mass spectrometry proteomics data have been deposited to the ProteomeXchange
Consortium via the PRIDE [1] partner repository with the dataset identifier PXD053717.



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
