# Peer review of "Part 2: Quantitative contributions of cyanobacterial alkaline phosphatases to biogeochemical"

_EGUsphere, 2024_

## Author Comment (AC3)

**Reviewer 1**

**General:**

The manuscript, No.: egusphere-2024-3996, Noelle A. Held et al., "Part 2: Quantitative contributions of cyanobacterial alkaline phosphatases to biogeochemical rates in the subtropical North Atlantic," integrates multiple methods to address questions about alkaline phosphatases in marine cyanobacteria and how these enzymes respond to nutrient limitation and different trace metals. The experiments and results appropriately address the questions posed by the authors about the abundance of these enzymes under different biochemical conditions and exposure to different metal cofactors. The experiments also revealed the unexpended finding of potentially promiscuous cofactor binding that remain open for future studies. The conclusions regarding biological complexity, the discussion of methodological caveats, and suggested future directions are appropriate for the outcomes of this current study. Overall, the design of the study, the method application of absolute quantitative proteomics, and the potential interest for the oceanographic and proteomics communities is good.

My main concerns after review are regarding the structure of the results section (details below) and whether this Part 2 manuscript is meant to be more of a methodological highlight, complementary to the *in situ* nutrient data and future climate discussion points in Part 1. Or, if the focus of both manuscripts is ultimately on the environmentally contextualized results (e.g., Figure 3). I would like the authors to comment on the specific comments/questions listed below:

We thank the reviewer for their constructive comments. This manuscript is intended to highlight both the methodological advancements and the biogeochemically contextualized results. While Part 1 focuses on the in situ/field results, this Part 2 manuscript focuses on the results of the incubation experiments and the interesting discrepancy between alkaline phosphatase responses in activity assays versus molecular assays. We have made major revisions to the text including reordering sections and adding a schematic to make this point more clear.

We address the specific comments below:

**Specific comments:**

If this Part 2 manuscript is meant to be an example application of "absolute quantitative proteomics" as a method, then it would benefit from a schematic methods comparison of the alkaline phosphatase rates in the "traditional" bioassay (Figure 1) vs. the strain-resolved approach (Figure 2).

We appreciated this idea but had a difficult time figuring out how to describe the difference between traditional bioassays and the proteomics assays in a schematic form. However we agree that this point can be highlighted more clearly. We have updated the abstract and introduction to highlight this point and the novelty of the combined biogeochemical-molecular approach used in the manuscript.

The alkaline phosphatase activity rates in the Figure 1 legend are also not clearly explained in the methods section.

We have added a methods section describing how these rates were measured and calculated.

- If the "significant responses" portion of the methods is the section that covers these rate calculations just spell that out a bit more.

  The title of this section has been modified to "Identification of significant responses to metal additions" for more clarity.

- If the rate calculations used in the "second to left column" in Figure 1 are based on the same principles used in the metalloproteome section that comes later regarding the absolute quantitative measurements then this should be made more clear. Either move the equations 1a - 2b to the methods and link them also to Figure 1 in the text or more clearly explain the rate calculations that went into Figure 1.

  The rates shown in Figure 1 (now Figure 2) are from the direct APA bulk assays. We have now added a section in the methods about how these assays are performed and the data analyzed. The rates calculations performed in Equations 1a-2b then use this data along with the proteomic data to estimate *Synechococcus*-specific hydrolysis rates and metal usage. These are distinct equations.

Results) Add a map of sampling stations to the "Biogeochemical setting" section. This would help orient the reader in the beginning of the Results section to understand the stations, environmental concentrations, and to which samples/stations the following assay results (Fig. 1 & 2) belong.

- This could be done by either adding a simple map of the sampling stations in addition to Table 3/Figure 1 or moving the section with the map (Figure 3 and associated text) up to the start of the Results section.

- Consider if the story works with Figure 3 and the APA discussion first and if not just add a simple station map before Figure 1.

This is a useful suggestion. We have reordered the manuscript sections and figures such that Figure 3 is brought up to the beginning of the manuscript as Figure 1. This provides a summary of the overall results which are then followed up with in more detail in later sections.

Figure 1) The main purpose of Figure 1 is to show that there were no significant differences in any of the "conventional parameters" measured at all four stations (i.e., a point of comparison for the significant differences in the next method presented) - this plot should be either condensed or simplified.

- A condensed Figure 1 should show only the Alkaline Phosphatase rate column at the four different stations, considering this is the more important comparison point for the next analysis in Figure 2 (leaving the other conventional parameters that are also not statistically significantly changing, Chl a and cell counts, to the text and supplement).

- If the bar plots are all kept together, simplify this plot by using a letter system (a – d) for the different parameters and incorporate this into the figure legend to make it easier to navigate than "left column" vs. "second to left column."

Thank you for this suggestion. The purpose of Figure 1 is to demonstrate the lack of significant differences in any of the traditional bioassays, and also to show that there is significant variation across replicates, which we interpret in light of past studies which show similarly patchy results, highlighting the need for novel approaches including the proteomic approach we explore in this manuscript. Given the importance of this point we would like to retain this data in the main text. We have added subplot annotations (letters) to this plot and replotted it (transposing the columns/rows) for clarity.

348) Are there any citations from model organism studies where they found multiple isoforms in the same organism?

Yes; a classic example is Bradshaw et al., 1981 which is among the first characterizations of E.coli alkaline phosphatase and identified at least three different isoforms differing by chemical modification. We have added the citation.

371) Not really an accurate use of "co-evolution," as this is a biological term typically defined by two species influencing each other's evolution. Be cautious with referring to chemical evolution and the biological evolution of a specific lineage of organisms in the same sentence. Maybe "the evolution of cyanobacteria in the dynamic chemical conditions of the ancient ocean…"

This is a fair point; we have revised to "simultaneous development of ocean chemistry and cyanobacterial metabolism"

430) Is there a citation for phosphorus stress studies? Unless it is covered with the Saito 2011b in the next sentence.

This is a speculative statement; we've revised it to clarify:

"One possible explanation for the presence of the many null responses across the basin is that organisms could be re-allocating metals towards use in alkaline phosphatases when under phosphorus stress."

The example in Saito 2011b provides a proof-of-concept example but does not describe this possibility specifically.

For the PRIDE data submission, add a sample description .csv file to the uploaded data that provides the MS file names and the corresponding stations and incubation conditions to help others navigate the data download for potential future reanalysis.

This has been added to the PRIDE submission.

**Technical corrections:**

275) add parentheses around the i.e. statement to match line 274

Corrected

308) provide the station coordinates or station name in parentheses after "westernmost"

Corrected

374) Watch out for changing between chemical symbols and full names. So far it has been very consistent, this was the only instance I spotted a full name.

Corrected; we also checked through the manuscript

411-417) Figure 3 legend: Citations are doubled.

Corrected (note this figure is now Figure 1). Seems to have been due to the citation manager.

420 – 424) Same citations having the doubling issue.

Corrected (this paragraph has been split up/re-written into the manuscript).

453) Supplement table referenced here should be Tables S3 and S4

Corrected.

In general check that the references to the supplement tables/figures are correct. Some of them seem to be switched around and supplement tables S5 -S7 and Figures S1-S5 are not referenced in the text. Last supplement figure should also be Figure S6. **Citation**: https://doi.org/10.5194/egusphere-2024-3996-RC1

Thank you for noticing this; these references have been corrected/added to the main manuscript and the supplement corrected.

**Reviewer 2**

Held and others study cyanobacterial alkaline phosphatases in the North Atlantic. The manuscript was well-written and well-referenced, and I only have a few minor comments to improve accuracy and readability.

Thank you for your comments and suggestions.

The abstract would benefit from numerical values

 We have added some numbers to the abstract.

Minor spacing errors, especially adjacent to references; note erroneous text on 125, double-reference on 420, etc. The manuscript needs to be carefully re-read for accuracy.

These issues have been corrected, particularly the double-references which originated from our reference manager and have now been removed.

193: lack of reference concerning but if there are no existing references this should be justified in more detail

Yes; unfortunately there is not yet a standard reference material for protein concentration in ocean samples, like there would be for many geochemical measurements. There is movement within the discipline to develop materials and improve intercomparison of proteomics data across experiments. Since reference materials are rarely used in proteomics in general and do not exist for environmental/ocean samples at all, we have removed this statement.

225: let the reader decide if the approach is accepted; it merely has been used before

 Fair point; corrected

Discussion points enter the results section, like in 303 'represents an interesting avenue for future research'. Is the results meant to be results and discussion? It certainly reads this way.

 Yes; we have relabeled this section as "Results and discussion"

The requirement for the reader to distinguish between red and green in Figure 3 needs to be reconsidered.

Thank you for this reminder; we have replotted this figure with a colorblind safe palette.

449: 'the PhoX isoform'

Corrected

460 and possibly elsewhere: don't use the * symbol to represent multiplication in formal mathematical equations, use the multiplication symbol.

Corrected throughout

497: 0.0347 nM h-1 is remarkably specific.

These numbers have been corrected to report with appropriate significant figures (2)

**Citation**: https://doi.org/10.5194/egusphere-2024-3996-RC2